# Evaluation of the Antimicrobial Activity of Cationic Peptides Loaded in Surface-Modified Nanoliposomes against Foodborne Bacteria

**DOI:** 10.3390/ijms20030680

**Published:** 2019-02-05

**Authors:** Stefania Cantor, Lina Vargas, Oscar E. Rojas A., Cristhian J. Yarce, Constain H. Salamanca, Jose Oñate-Garzón

**Affiliations:** 1Grupo de investigación en Química y Biotecnología (QUIBIO), Facultad de Ciencias Básicas, Universidad Santiago de Cali, Calle 5 No. 62-00, Cali 760035, Colombia; estephye@hotmail.com (S.C.); lina-vargas29@hotmail.com (L.V.); oerojas@usc.edu.co (O.E.R.A.); 2Laboratorio de Diseño y Formulación de Productos Químicos y Derivados, Departamento de Ciencias Farmacéuticas, Facultad de Ciencias Naturales, Universidad Icesi, Calle 18 No. 122-135, Cali 760035, Colombia; cjyarce@icesi.edu.co

**Keywords:** Cationic antimicrobial peptide, polymer-coated liposomes, foodborne pathogens

## Abstract

Bacteria are a common group of foodborne pathogens presenting public health issues with a large economic burden for the food industry. Our work focused on a solution to this problem by evaluating antibiotic activity against two bacteria (*Listeria monocytogenes* and *Escherichia coli*) of relevance in the field of foodstuffs. We used two approaches: *(i)* structural modification of the antimicrobial peptides and *(ii)* nano-vehiculisation of the modified peptides into polymer-coated liposomes. To achieve this, two antimicrobial peptides, herein named ‘peptide +2′ and ‘peptide +5′ were synthesised using the solid phase method. The physicochemical characterisation of the peptides was carried out using measurements of surface tension and dynamic light scattering. Additionally, nanoliposomes were elaborated by the ethanol injection method and coated with a cationic polymer (Eudragit E-100) through the layer-by-layer process. Liposome characterisation, in terms of size, polydispersity and zeta potential, was undertaken using dynamic light scattering. The results show that the degree of hydrophilic modification in the peptide leads to different characteristics of amphipathicity and subsequently to different physicochemical behaviour. On the other hand, antibacterial activity against both bacteria was slightly altered after modifying peptide sequence. Nonetheless, after the encapsulation of the peptides into polymer-coated nano-liposomes, the antibacterial activity increased approximately 2000-fold against that of *L. monocytogenes*.

## 1. Introduction

Foodborne illnesses have significant impacts on global healthcare systems. There is a growing concern regarding foodborne bacterial pathogens; the World Health Organisation (WHO) estimates that, in 2010, foodborne illnesses affected 600 million people globally and caused 420,000 deaths [1]. To contribute to solving this public health problem, alternative antibiotics have emerged, such as antimicrobial peptides (AMPs); e.g., Nisin is an AMP widely used as an additive to foodstuffs and is the only bacteriocin approved by FDA and European Union (E234) [2]. AMPs are natural antimicrobial agents constituting the innate immune system, which mainly target anionic microbial membranes as consequence of peptide cationic charge, a result of the initial interaction between the peptide and the microbial membrane [3,4]. Subsequently, the hydrophobic amino acids are inserted into the hydrophobic core of the membrane, increasing the disorder of the phospholipids while the barrier membrane function is lost [3]. Thus, other peptide structural properties, such as hydrophobicity, amphipathicity and secondary structure, play a pivotal role to achieve insertion into the microbial membrane [5]. Indeed, such structural properties can modulate the physicochemical characteristics of both the peptide and the antimicrobial activity [6,7].

Alyteserin-1c is an AMP comprising a 23 amino acid residue (net charge: +2) and was first isolated from norepinephrine-stimulated skin secretions from the midwife toad *Alytes obstetricans* [8]. Alyteserin-1c has shown antibacterial selectivity against Gram-negative bacteria, exhibiting a MIC of 25 µM for *E. coli*. Furthermore, low hemolytic activity against human erythrocytes has been described [8,9]. Although AMPs are an alternative means for eliminating bacteria in foods, due to their biocompatibility, biodegradability, broad spectrum of activity and potent bactericidal properties, most have not been successfully applied to food preservation, since their susceptibility to enzymatic degradation limits their bioavailability [10]. Efforts to improve peptide stability against a range of environmental and chemical stresses have been developed, for instance vehiculisation using liposomes composed of membrane phospholipids exhibiting excellent biocompatibility [11,12,13]. However, such systems are thermodynamically unstable, resulting in aggregation phenomena by mechanisms such as flocculation and coalescence during storage that can lead to a burst release of the active agent [14]. Thus, the liposomal surface can be decorated with polymers to enhance its stability and functionality. Eudragit E-100^®^ is a polymer widely used for enteric coating, categorised as nontoxic and nonirritant and is approved by the FDA Inactive Ingredients Guide (in their forms of oral capsules and tablets) [15]. It is therefore an inexpensive polymer with potential utility to the food industry for coating liposomes. 

Recent studies with Nisin encapsulated into polymer coated-liposomes have shown that it exhibits an increased peptide stability and antimicrobial activity against foodborne pathogens. Niaz et al. [2] found that Nisin-loaded, multi-component colloidosomes have superior potential to control resistant foodborne pathogens when compared to free Nisin. Additionally, Nisin-loaded pectin or polygalacturonic acid-coated liposomes exhibited an enhanced efficiency against *L. innocua 6a* when compared with free Nisin [16]. Thus far, there have been few studies examining the antimicrobial effect of AMPs encapsulated into coated liposomes [16,17,18,19] and the current knowledge of how the structural and physicochemical properties of peptide impacts their ability to be encapsulated into liposomes is limited.

The aim of this research was to design and synthesise a peptide (+5) from Alyteserin-1c (+2) by the replacement of hydrophobic amino acids by hydrophilic amino acids at the polar face of the helix, increasing both the amphipathicity and cationic charge while decreasing hydrophobicity. Analyses of the structural prediction and physicochemical properties of each peptide in solution were performed. Eudragit-coated liposomes were used as system encapsulating peptides, and the effect of the structural and physicochemical properties of the peptides on encapsulation and biological activity were reported. Susceptibility tests using encapsulated and free peptide were performed against foodborne bacteria. 

## 2. Results and Discussion

### 2.1. Peptides Design and Sequence Characteristics

The peptide +2 (H0USY4, code UniProt KB), constituting 23 residues, is Alyteserin-1c isolated from the amphibian *Alytes obstetricans* and has a sequence described by Conlon et al. [8]. Peptide +2 was selected as a template sequence due to both, its reduced positive net charge and the presence of hydrophobic amino acids in the polar face of the helix, in order to explore the effect of increasing charge and hydrophilicity in that helix face on the biological activity, encapsulation ability and physicochemical properties. Peptide +5 is a derivate of the peptide +2, which has increased cationic properties as a result of the rational replacement of anionic and hydrophobic residues by hydrophilic and cationic residues at the polar face, shown in Table 1 (bold letters), following the Bordo and Argos suggestions [20] and maintaining similar structural properties after substitutions. The characteristics of both peptides are summarised in Table 1, including the hydrophobic character, amphipathicity, molecular weight and the net charge at pH 7.4. Peptide +5 had four substitutions (E4R, A8S, S12K and A18S). Furthermore, the substitution of hydrophobic alanine for hydrophilic serine at the polar face decreased its hydrophobicity from 0.461 (peptide +2) to 0.373 (peptide +5), whereas the hydrophobic moment was increased from 0.380 (peptide +2) to 0.434 (peptide +5) (Table 1). Evidently, altering one structural property will often result in significant changes to one or more of the other properties.

After the substitution of these residues, the modified peptides exhibited enhanced hydrophilicity in the polar face of the helix, whereas the hydrophobic face was unaltered (Figure 1B). At the first 18 amino acids of the sequence, the polar face consisted of 10 amino acids, of which 6 were conserved in the modified peptide (G1, K3, K7, G11 and K15). On the other hand, the hydrophobic face is constituted by 8 amino acids: one F, two I, three L and two G (Figure 1).

### 2.2. Molecular *Dynamic Simulation*

Comparing both models, peptide +5 and peptide +2, after energy minimisation shows that the transition of each residue towards minimum energy was mainly in the side chain, without further geometric optimisation of the amine (-NH) and carboxyl (-COOH) functional groups. Each residue showed a spatial shift, which was calculated by analysing the distances (Å) between the last functional group of the side chain of each residue of the peptide +5 in comparison with peptide +2, where the residue that showed the greatest displacement to achieve the zero-energy trend was Lys12, calculated by comparing the nitrogen Z atom at a distance of 1.49 Å (Figure 2). A greater displacement tendency was evidenced after the energy minimisation of peptide sequence residues from Lys12 to Ser23, where displacements with distances between 0.97 Å and 1.49 Å were applied.

The molecular dynamic (MD) simulation was performed using Periodic Boundary Conditions that are effective in eliminating surface interactions with water molecules and creating a more faithful representation of the in vivo environment [21]. The simulation time performed was sufficient to capture the long-time dynamics of a simple bimolecular system that required only the analysis of changes in the secondary structure and the determination of electrostatic interactions with the solvating model. After molecular dynamics, a NH-terminal sequence (GLKRIFKSGLGK) of peptide +5 interacting in a water box is responsible for the highest electrostatic interactions compared with the COOH-terminal region (Figure 3A). Arg4 is responsible for the electrostatic potential grid border and Lys15 exhibits electrostatic interaction only through its nitrogen atom, since these atoms in the side chain are more available for creating the electrostatic potential grid.

During the first 500 ps of molecular dynamics, the peptide +5 lost α-helix structure at the amino acid sequence AHVAS (Figure 3A). The replacement of alanine 18 by serine appears to contribute to the loss of the α-helix structure since alanine has the highest helix-forming propensity [22]. Special attention was required in the case of residue Arg4 because, throughout all molecular dynamic simulations, it showed high movement in its side chain (Figure 3B) owing to the reactivity of the guanidine group. This side chain plays a pivotal role in microbial membrane interaction since it can approach the heads of phosphatidylglycerol to a distance of 5 Å [23]. Furthermore, arginine has the ability to internalise the peptides into the cells by crossing the membrane [24].

### 2.3. Characterisation Physicochemical of Peptides in Aqueous Media

The results of the surface tension and aggregation index with respect to the concentration of the peptides +2 and +5 are shown in Figure 4.

The results show a marked effect depending on both the medium (ultrapure water or phosphate buffer) and the degree of hydrophilicity of the peptide face. Regarding the medium, when the peptides are dissolved in ultra-pure water, it is observed that there are variations neither in surface tension nor in the aggregation index. This result is very interesting and leads us to propose two possible hypotheses. (i) When the peptides are in the water, these could be mostly adsorbed on the container surface and thus, the peptide amount dissolved in ultra-pure water is insufficient to give some response in regards to the methodologies employed. (ii) There is an effect provided by the hydrophilic character in the peptide face. This result can be explained by the hydrophilic character of both peptides like that, being in an aqueous medium (free ions), they tend to be completely solvated in the bulk of the system. On the contrary, when the peptides are in a phosphate buffer medium, the cationic counter ions (from the buffer) cause a charge shielding effect on the peptide face. This effect generates a change in the hydrophilic-hydrophobic character of the peptide structure, passing from a hydrophilic to an amphipathic structure and, thus, changing the thermodynamic behaviour similarly to a surfactant. Comparing the results of both peptides, it is observed that in the peptide +5 (more hydrophilic), changes in surface tension and aggregation index are achieved at lower concentrations (5–10 µM) than with peptide +2 (20 µM). This result suggests that peptide +5 acquires a more amphipathic character than peptide +2 and that peptide +5 tends more to aggregation. These results are significant as it has been described that peptide or protein systems are biologically more active when they are in non-aggregated form [25,26]. These results are necessary to project the vehiculisation process within the liposomes, where the peptides could interact with the components forming the lamellar structure, affecting nano-formulation.

### 2.4. Polymer-Coated Liposome Coated with Peptides

The results of the DLS characterisation of the polymer-coated liposomes, both free and loaded with the peptides +2 and +5, are shown in Figure 5.

Figure 5A shows that the non-coated and unloaded peptide liposomes (NCL-F) have sizes around 235 nm, however, when they are loaded with the peptides +2 (NCL-peptide + 2) and +5 (NCL-peptide + 5), respectively, their sizes increase to almost twice their initial size, with a marked increase in polydispersity index (PDI) (Figure 5C) from low polydispersity (PDI < 0.3) (NCL-F) to high polydispersity (PDI 0.5–0.7) (NCL-peptide +2 and +5). This result can be explained by considering several aspects: *(i)* the peptides are located inside the liposome in the internal aqueous compartment, *(ii)* the peptides are located on the liposomal surface, or *(iii)* the peptides are located on both sides. According to the zeta potential results (Figure 5B), such values tend to be negative due to the nature of the components used to form the liposomal structure (NCL-F). On the contrary, when the peptides are present, the zeta potential values become less negative, suggesting that some amount of cationic peptide could be interacting with the surface. Therefore, it can be determined that both peptides are located both inside the liposome and in the lamellar structure.

Conversely, the liposomal coating process shows changes in size, polydispersity and zeta potential. Size increases tend to be proportional both in the unfilled liposomes (CL-F) and in those loaded with peptides (CL-peptide +2 and +5). The polydispersity index increases in all cases, suggesting that the coating polymer adheres in different ways to the liposomal surface, leading to different size populations.

In relation to the zeta potential results, an inversion in the sign of the values from negative to positive (CL-F) is shown, suggesting that the coating polymer effectively adheres to the polymeric surfaces. The aforementioned result, which positive zeta potential is coherent if it is considered that the polymeric material used corresponds to a cationic salt derived from the Eudragit E-100 and, therefore, it could provide such a surface. Furthermore, as the charge on the peptide face increases, the zeta potential values become higher (CL-Peptide +2 and +5), thus reaffirming that the peptides are distributed both inside the liposome and in the lamellar structure (Figure 5D).

Conversely, it should be mentioned that although several methodologies were attempted to quantify the degree of encapsulation of the peptides within the nano-liposomes, such as dialysis, pressure-assisted ultrafiltration, ultrafiltration/centrifugation, but none were successful. This was because the ammonium salt present in the polymer derived from Eudragit E-100 interfered with the peptide quantification methodology (Bradford colorimetric method). To achieve such quantification, it would be necessary to employ a discriminative and sensitive methodology, such as mass spectrometry coupled to liquid chromatography with UV-Vis detector. However, the results nonetheless indirectly suggest the vehiculisation of the peptides in the liposomes but the peptide encapsulated fraction is now and interesting topic to be addressed in future works.

### 2.5. Stability of Liposomes

The results of the stability study of the non-coated liposome (NCL) and polymer-coated liposome (CL) showed that the liposome surfaces modified with the Eudragit E-100 polymer were more stable than the non-coated liposomes (Figure 6). This result is consistent with the electrostatic stablisation effect, which increases with the adsorption of the polymer on the liposome surface and forms a positive electrostatic film preventing liposomal aggregation and extending the limits of physical stability. This result is very interesting because it shows that coated liposomes can reach a high stabilisation. However, it is important to highlight that to give a greater significance to this result, it will be necessary to carry out new comparative studies against other types of cationic liposomes (no-coated), such as the glutathione disulfide liposomes, which have shown very interesting characteristics corresponding to low toxicity, bioability and adequate physico-chemical stability [27,28,29].

### 2.6. Antimicrobial Activity

Antibacterial activity against Gram-negative bacteria was determined, showing that peptide +2 exhibited increased activity, unlike peptide +5, reaching a MIC value of 15.2 μM for *E. coli* whereas peptide +5 exhibited a MIC of 62.5 μM. In Gram-negative bacteria, the self-promoted uptake pathway is a proposed mechanism that appears to be related with bacterial death, where the cationic peptide displaces divalent cations associated with lipopolysaccharides (LPS), destabilising the macromolecular complex and promoting the internalisation of the peptide to the inner membrane [30]. However, the reduced activity exhibited in peptide +5 could also be due to the impediment of crossing the outer Gram-negative membrane to reach the periplasmic space and inner membrane, as a consequence of reduced hydrophobicity when replacing Ala/Ser at the polar face [5]. Hydrophobicity is thus an important parameter for peptide antibacterial activity as it controls the extent to which the peptide can partition into the hydrophobic core of the membrane [3]. Furthermore, the loss of the helical structure in the COOH-terminal region of peptide +5 (Figure 3A) affects its antibacterial activity, since the disruption of the helical structures can lead to a dramatic decrease in activity [5]. On the other hand, the encapsulation of peptides into NCL had no additional antimicrobial effect against *E. coli*, since these colloidal systems can be degraded by flocculation or coalescence processes [14]. Nonetheless, after coating the liposome with Eudragit polymer, the MIC was reduced, reaching values of 1.25 and 5 μM for CL-peptide +2 and CL-peptide +5, respectively, corresponding to an increase in antibacterial activity of the peptide +2 (peptide+2/CL-peptide +2) of approximately 12.5 times after being encapsulated into the liposomal formulation. These vehicles would avoid chemical degradation by bacterial proteases of the encapsulated peptides, and thus, their release onto the bacterial surface is favoured [31]. After the CL is anchored to the bacterial surface, Eudragit would be stripped of liposome and NCL is internalised in the cell by endocytosis (or phagocytosis), followed by the enzymatic digestion of the liposome in the intracellular compartment (endosome, phagosome or acidosome) [32], while the intact AMP is distributed over the bacterial membrane.

Conversely, increasing the hydrophilicity, charge and amphipathicity of the Alyteserin-1c peptide sequence resulted in a slight enhancement of antimicrobial activity in *L. monocytogenes,* with MIC values of 125 and 62.5 μM for peptide +2 and +5, respectively. The cationic charge has been widely associated with the antimicrobial activity of AMPs [6,33,34] since the surface charge density of the membrane determines the magnitude of the electrostatic attraction (Coulombic) linking the positively charged molecules of the peptide to the negatively charged lipid membranes [35]. Thus, peptide +5 can be more strongly attracted by anionic groups positioned toward the outside of the cell wall than its analogue +2, due to the presence of carboxyl groups of the muramyl peptides of peptidoglycan and the carboxyl and phosphate groups of teichoic acids located at Gram-positive external envelope [36]. Furthermore, bacterial anionic membranes constituted by anionic phospholipids [3] also contributes to the electrostatic attraction of cationic peptides.

After encapsulating peptides into NCLs, changes in the antibacterial activity against *L. monocytogenes* were not observed, a result consistent with previous observations in *E. coli*. However, after coating the NCL with polymer, an increase in activity against *L. monocytogenes* was observed, reducing the MIC to 0.06 and 0.04 μM for CL-peptide +2 and CL-peptide +5, respectively. It is important to mention that a part of this activity is provided by the blank coated liposome (CL-F), which exhibited a MIC of 3.9 μM, contrasting with that observed in *E. coli* where such formulation did not exhibit any antibacterial effect at the maximum tested concentration. Previous studies have shown that Eudragit shows a membrane-destabilising effect that is expressed as a perturbation of the membrane structure allowing the passage of water-soluble molecules, and also antiviral effects against the herpes simplex virus type 2 (HSV-2) [37]. Formulations with traditional antibiotics have also been described using Eudragit to increase their activity against *P. aeruginosa* with fluoroquinolone resistance [38]. Therefore, the encapsulation of the peptide +2 and the peptide +5 into CL increases the activity by 2083 (peptide+2/CL-peptide+2) and 1562 times (peptide+5/CL-peptide+5) against *L. monocytogenes*. Excluding the antibacterial contribution of CL-F, peptides +2 and +5 encapsulated into CL exhibited an enhanced activity of 65 and 97.5 times more than the unencapsulated peptides, respectively. In spite of that, the target of AMPs are the microbial membranes, they can be encapsulated and transported into membrane models without breaking them, since these models contain cholesterol and metastable zwitterionic phospholipids, which contribute to physical stability compared to a bacterial membrane [39,40].

## 3. Materials and Methods

### 3.1. Bacterial Strains and Chemicals

*L. monocytogenes* ATCCbaa751 and *E. coli* ATCC25922, were obtained from the American Type Culture Collection (ATCC; Rockville, MD, USA). NovaPEG Rink Amida resin, Fmoc-protected amino acids, 2-(1H-benzotriazole-1-yl)-1,1,3,3-tetramethyluronium tetrafluoro-borate (TBTU), Piperidine, N,N-Diisopropylethylamine (DIEA), Ninhydrin, Dimethylformamide (DMF), Trifluoroacetic acid (TFA), 1,2-Ethanedithiol (EDT), Triisopropylsilane (TIS) and Mueller Hinton Broth (MHB) were purchased in Merck (Darmstadt, Germany). Cholesterol, Epikuron 200^®^ and dioleoyl-phosphatidyl-ethanolamine (DOPE) were obtained from Avanti Polar Lipids (Alabaster, Alabama, USA). Eudragit^®^ E-100 was purchased in Evonik (Darmstadt, Germany). Ampicillin (Fersinsa Gb) was supplied by Tecnoquímicas S.A. (Cali, Colombia) and Gentamicin solution (GENFAR^®^, Cali, Colombia) was purchased from a local pharmacy. The datasets used and materials using during this study are available from the corresponding author on reasonable request.

### 3.2. Peptide Design

A sequence of the helical template peptide Alyteserin-1c (charge +2, PDB code 2L5R), comprising 23 amino acid residues, was obtained from Conlon et al., [8]. Four amino acid substitutions were introduced in the polar face of amphiphilic helix into the wild-type (+2) sequence, following the suggestion of Bordo and Argos [20], in order to generate an analogue peptide (+5). The net charge of each peptide after residue substitution was calculated as the addition of the basic residues. Peptide hydrophobicity <H>, defined as the mean hydrophobicity value from all residues within a peptide according to a standard scale [41], and the hydrophobic moment <µH>, a quantitative measure of the amphipathicity defined as the vectorial sum of individual amino acid hydrophobicity vectors normalised to an ideal helix [42], were calculated using Heliquest software (http://heliquest.ipmc.cnrs.fr/) (access on 6th march 2018). The structure prediction of peptide +5 was performed using a homology-based modelling procedure, where the amino acid sequence of Alyteserin-1c was the template for the amino acid substitutions, and side-chain modeling was carried out using DeepView Swiss-PdbViewer software (http://www.expasy.org/spdbv/) (access on 25th may 2018).

### 3.3. Peptide Synthesis

The syntheses of Alyteserin-1c and its analogue were performed by solid phase methodology using NovaPEG Rink Amida resin and microwaves for coupling. After swelling, deprotection and washing of the quantified resin, a solution of activators (TBTU and DIEA) was mixed with Fmoc-protected amino acids. This mixture was irradiated with microwaves and washed with 20% piperidine in DMF for Fmoc deprotection. Deprotection and coupling were repeated with sufficient Fmoc-protected amino acids to obtain a peptide-resin complex. After each coupling step, the Kaiser test was used in order to confirm completeness of the coupling. Subsequently, the resin was washed twice successively with ether and was dried at 37 °C. Peptide was cleaved from the resin by treatment with Method 2 (94% TFA/2.5% water/2.5% EDT/TIS 1%) for 3 h under shaking conditions at room temperature, then dried and precipitated with cold ethyl ether. The ethyl ether was removed by centrifugation at 5000 rpm. Following this, peptides with 95% purity were obtained by reverse phase, high-resolution liquid chromatography using a semi-preparative Chromolith^®^ RP-18e column and applying a mixture of: (A) H_2_O with 0.05% TFA (*v*/*v*) and (B) acetonitrile containing 0.05% TFA (*v*/*v*) as the mobile phase. For the elution of peptides, a programmed gradient of 70 min with 0 × 50% B at 1 mL × min^−1^ and detection at 220 nm was used. The purity was verified using MALDI-TOF mass spectrometry (Bruker Daltonics, Bremen, Germany; MALDI Biotyper).

### 3.4. Molecular Dynamic Simulation

Electrostatic interaction potential was calculated using periodic boundary conditions (PBCs) and Particle Mesh Ewald (PME) methods in explicit solvent [43]. A TIP3 implicit water model simulation [44] with a 5 Å rigid cubic cell was used to measure from the atom with the largest coordinate in every direction, (Vector 1: 31 Å, Vector 2: 44 Å and Vector 3: 26 Å). A total of 2472 atoms were inserted, corresponding to 824 water molecules, with a final confirmation of: 2832 atoms, 2009 chemical bonds, 1485 angles, 950 dihedral angles, 2665 rigid bonds and 991 hydrogen groups. The structure (PDB file) and coordinates (PSF file) for Alyteserin-1c (2L5R) used for mutation were generated from topologies and initial force field parameters obtained from the Protein Data Bank (RCSB). The initial orientation was attained with Swiss.pdb Viewer. CHARMM force fields [21] were used throughout. To include long range electrostatic interactions in the simulations, the particle mesh Ewald method for periodic systems was used [43], since it creates a 3D grid over which the charge is distributed. The grid size was configured to 1 Å. The energy of the Peptide model was minimised every 100 steps with a 12 Å cutoff. The MD simulation visualisation file was set to 250 steps per visualisation. The simulation time was 2000 ps with 2 fs per step (1000000 steps). Temperature and pressure were 310 K and 1 atm, respectively, achieving an isobaric-isothermal ensemble. Visual Molecular Dynamics (VMD1.9.3, for visualisation) and the Scalable Molecular Dynamic (NAMD2.12) simulation software were used [45].

### 3.5. Physicochemical Characterisation of the Peptide in Aqueous Media

#### 3.5.1. Surface Tension Measurements

Surface tension measurements of the peptides were carried out using a contact angle meter (OCA15EC Dataphysics Instruments, Filderstadt, Germany) with a software driver (version 4.5.14 SCA22), where the data capture was recorded using an IDS video camera. For this, the pendant-drop method [46,47] was performed in triplicate, where peptide concentrations between 0 and 50 µM were prepared with ultrapure water and phosphate buffer saline pH 7.2 (PBS, 138 mM NaCl, 3 mM KCl, 1.5 mM NaH_2_PO_4_, 8.1 mM Na_2_HPO_4_).

#### 3.5.2. Aggregation Index Measurements

The samples of peptides +2 and +5 were prepared in serial concentrations between 0 and 50 µM in PBS buffer pH 7.2 at 138 mM of ionic strength. An aliquot was analysed immediately after preparation. Dual angle DLS measurements (173° and 13°) of automatic duration were carried out using a Zetasizer Nano ZSP (Malvern Instruments, Worcestershire, United Kingdom). The Aggregation Index is a parameter based on the mean z-average size measured for the two angles of scattering according to the equation:
(1)$$\mathrm{Aggregation}\;\mathrm{index}=[\frac{Z_{\mathrm{fwd}}}{Z_{\mathrm{bkd}}}-1]$$
where, *Z_fwd_* is the average particle size detected by forward scatter optics at 13°. Trace amounts of aggregate are often highly useful for the enhanced detection of large protein aggregates, whereas *Z_bkd_* is the particle size average at a detection angle of 173° (non-invasive back-scatter).

### 3.6. Polymer-Coated Liposome Coated with Peptides

#### 3.6.1. Preparation of Liposomes Loaded with Peptide

Liposomes were developed using the ethanol injection method. The liposomes were prepared on the basis of a sequential process defined in several steps. Step 1, (preparation of organic phase): ethanolic solutions of Epikuron 200™ (1.3 mg/mL), cholesterol (0.64 mg/mL) and DOPE (1.23 mg/mL) were prepared, from which volumes of 42.3, 42.4, and 15.3 µL were taken, respectively, to obtain 100 µL of the lipid mixture and settled in a 200 µL eppendorf PCR chemically inert, prime virgin polypropylene (PP) tube (Eppendorf AG, Hamburg, Germany). Step 2 (phase mixture): 100 µL of organic phase was slowly injected to 100 µL of aqueous phase composed of peptides at a concentration of 125 μM dissolved in phosphate buffered saline (PBS, 138 mM NaCl, 3 mM KCl, 1.5 mM NaH_2_PO_4_, 8.1 mM Na_2_HPO_4_, pH 7.2) which were stirred (in vortex) for 1 min and left under ‘aging’ by 10 min. Step 3 (formation of liposomes): The resulting mixture between the organic and the aqueous phases were diluted in 300 µL of the respective aqueous media. Step 4 (liposome purification): The diluted mixture was centrifuged at 10,000 rpm in a micro-centrifuge Hettich RCF 10,538 (Andreas Hettich GmbH and Co.KG, Tuttlingen, Germany) for 6 min, using 500 µL centrifugal filters tubes (VWR, Radnor, PA, USA) with a pore size of 30 kDa. Subsequently, the fractions of purified liposomes (supernatant of filtering process) were extracted, resuspended and adjusted to a volume of 1000 µL in the respective aqueous media.

#### 3.6.2. Liposome Surface Modification

An aqueous solution of Eudragit^®^ E-100 (0.7% w/v) adjusted to pH 4.0 with 0.1 M HCl, was added to liposomal dispersion loaded with peptide (previously elaborated), at a ratio of 1:1 and at a rate of 50 µL/min. Subsequently, the mixture was left under constant magnetic stirring at 300 rpm for 8 h in a polypropylene closed vessel. Finally, it was centrifuged at 10,000 rpm for 2 min, using the centrifugal filters with 30 kDa cut-off. This surface modification technique was previously reported and it is called layer-by-layer coating process [48].

#### 3.6.3. Physicochemical Characterisation of Liposomes

Particle size and zeta potential were determined using a Zetasizer nano ZSP (Malvern Instrument, Worcestershire, United Kingdom) with a red He/Ne laser (633 nm). Particle size was measured using dynamic light scattering (DLS) with an angle scattering of 173° at 25 °C, in a quartz flow cell (ZEN0023), whereas zeta potential was measured using a disposable folded capillary cell (DTS1070). This instrument reports particle size as the mean particle diameter (z-average), and PDI ranging from 0 (monodisperse) to 1 (very broad distribution). All measurements were performed in triplicate after an appropriate dilution (5:5000, *v*/*v*) of the liposome suspension in ultra-pure water and were reported as the mean and standard deviation of measurements made from freshly prepared liposomal dispersions.

#### 3.6.4. Stability of Liposomes

The stability of the coated and non-coated liposomes was assessed using a stability chamber at 40 ± 1 °C, where the change in liposomal size was evaluated for 7 days in triplicate.

### 3.7. Antimicrobial Activity

Microbial susceptibility tests were performed according to clinical and laboratory standards institute (CLSI) standard methods [49]. Bacteria were inoculated in MHB and incubated overnight at 37 °C. The culture was then diluted in MHB until an OD_625_ of 0.1 was reached (approximately 1 × 10^8^ CFU/mL) and additional incubation was continued for 30 min. Such cultures were diluted by a factor of 1:200. Subsequently, 90 μL of bacterial culture was incubated for 18–20 h into 96-well plates at 37 °C with 10 μL of free or encapsulated peptide until a final inoculum of approximately 5 × 10^5^ CFU/mL was reached. Treatments were applied at different serial concentrations ranging from 15.2 to 250 μM. As negative control, PBS was used, and for positive control Gentamicin/ampicillin were used for Gram-negative and Gram-positive bacteria, respectively. After incubation, the minimum inhibitory concentration (MIC) was determined by visual analysis.

### 3.8. Statistical Analysis

Biological and physicochemical assays were performed in triplicate. After confirming the assumptions of normality and variance homogeneity, the data were analysed using ANOVA and Tukey’s multiple comparisons method to determine if there are significant differences between the means obtained, using a significance level of *p* < 0.05. All analyses were performed using the Statgraphics Centurion XVI software (StatPoint Technologies Inc, Warrenton, Va, EE. UU.).

## 4. Conclusion

The purpose of the replacement of amino acids in this study was to develop an approach to an ideal amphipathic α-helix with hydrophobic residues at one side of the helix and cationic/hydrophilic residues at the other side. This involved the introduction of hydrophilic amino acids at the polar face. We found that hydrophobicity/amphipathicity and charge have effects on the physicochemical properties of both the peptides and the liposomes encapsulating them. Additionally, by substituting amino acids, an increase of specificity toward Gram-positive bacteria can be achieved. On the other hand, the encapsulation of the peptide Alyteserin-1c into polymer-coated liposomes increased the antibacterial activity significantly (2083 times) against *L. monocytogenes* and modestly (12.5 times) against *E. coli*, in comparison with the unencapsulated peptide. Both bacterial strains present a serious risk when consumed in contaminated food, since they are extremely virulent in humans. *L. monocytogenes* has particularly remarkable resistance to the usually deleterious effects of freezing, drying and heating [50]. Therefore, the application of funtionalised liposomes coated with food-safe polymers for the encapsulation of small traces of biocompatible non-traditional antibiotics results in a potent bactericidal formulation and provides a promising solution to mitigate the bacterial resistance problem toward food storage conditions, avoiding the spread of infections by foodborne pathogens.

## Figures and Tables

**Figure 1 ijms-20-00680-f001:**
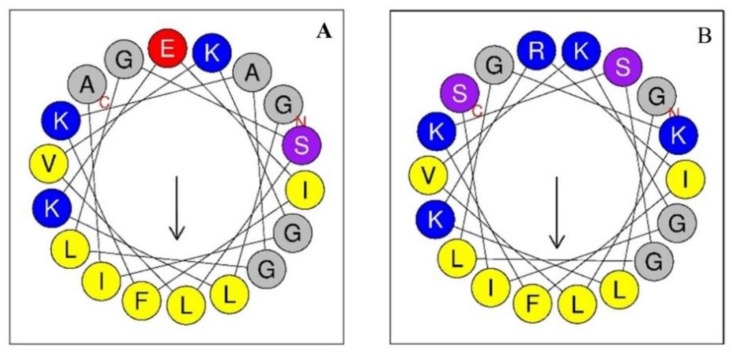
Wheel projections of the first 18 residues of the sequence of each peptide. (**A**) Alyteserin-1c peptide (+2); (**B**) peptide +5. The hydrophobic amino acids are yellow, and the charged amino acids are blue (net positive) or red (net negative). The polar amino acids are purple and those in-between are grey.

**Figure 2 ijms-20-00680-f002:**
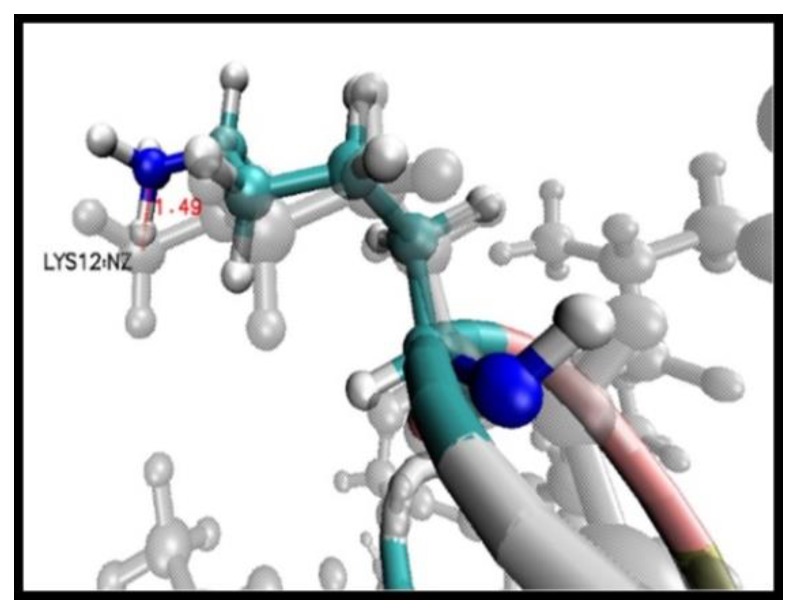
Atomic representation of the amino acid Lys12 in peptide +5 (color) compared with peptide +2 (gray); the nitrogen atom (blue color) is displaced 1.49 Å to achieve energy minimisation.

**Figure 3 ijms-20-00680-f003:**
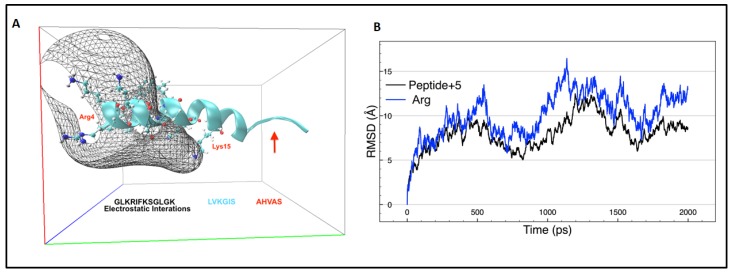
(**A**) Smoothed electrostatic potential grid model taken during 2000 ps of molecular dynamics simulation and characterisation of the residues and sequence involved in the electrostatic interaction. (**B**) Root-mean-square deviation (RMSD) against time in picoseconds (ps); the black line represents the behaviour of all residues and the blue line represents the behaviour of only Arg4.

**Figure 4 ijms-20-00680-f004:**
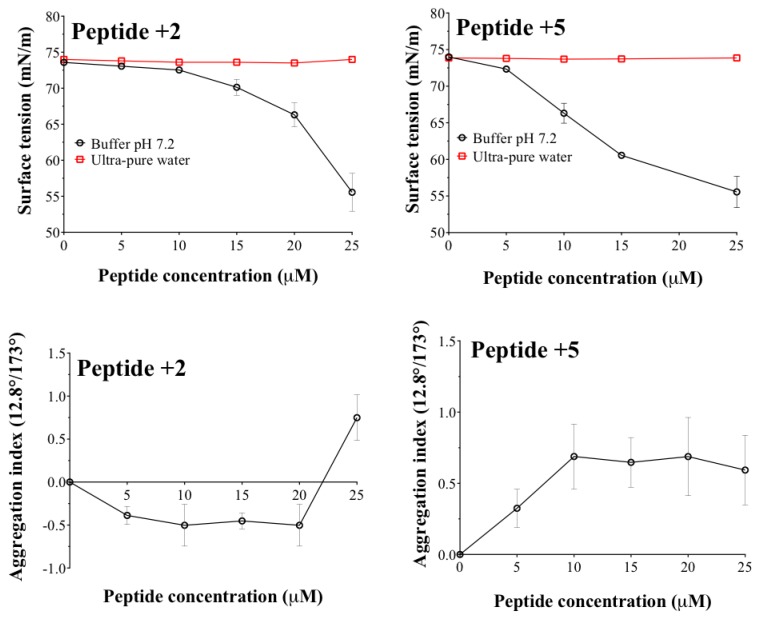
Changes in aggregation index and surface tension as a function of peptide concentration.

**Figure 5 ijms-20-00680-f005:**
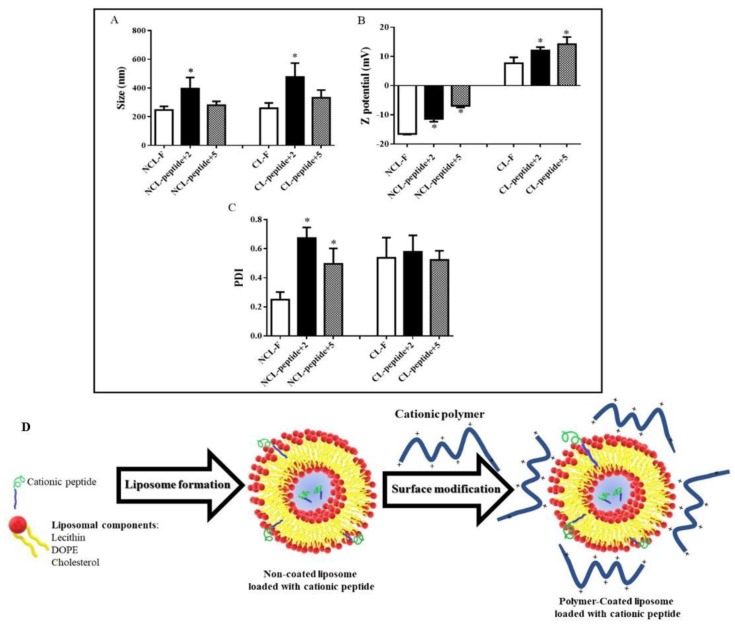
Mean values of (**A**) particle size, (**B**) Z-potential, (**C**) polydispersity index (PDI) and (**D**) scheme formation of liposomes loaded with cationic peptides +2 and +5 before (NCL) and after (CL) the coating process. Data are the average of at least three independent experiments ± s.d. Error bars represents the standard deviation. * Significant difference *p* < 0.05 to the NCL and CL without peptide (NCL-F and CL-F, respectively).

**Figure 6 ijms-20-00680-f006:**
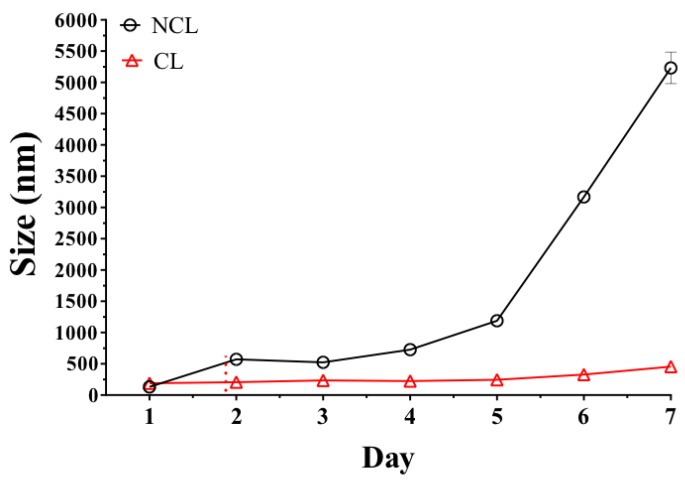
Evaluation of liposomal size change with time.

**Table 1 ijms-20-00680-t001:** Peptide sequences and properties.

Name	Sequence	Q	<H>	<µH>	MW
	1 10 20				
peptide +2	H_2_N- GLKEIFKAGLGSLVKGIAAHVAS–COOH	+2	0.461	0.38	2266.7
peptide +5	H_2_N- GLK**R**IFK**S**GLG**K**LVKGI**S**AHVAS–COOH	+5	0.373	0.434	2366.9

*Q = charge, <H> = hydrophobicity, <µH> = amphipathicity, MW= Molecular weight (g/moL).*

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
