# Peer review of "Evaluation of the Antimicrobial Activity of Cationic Peptides Loaded in Surface-Modified Nanoliposomes against Foodborne Bacteria"

_ijms, 2019, doi:10.3390/ijms20030680_

Reviewer 1 Report

Researchers of this manuscript developed new Anti-microbial peptides and used E-100 coated nano-liposomes to improve stability of food.  This is a well written manuscript. 

1. How did eudragit E-100 prevented agglomeration.  If it is a positive-positive repulsion between nano-liposomes, then why cant researchers make cationic liposomes instead of coating liposomes with E-100.  Please cite (In-vitro and in-vivo tumor growth inhibition by Glutathione disulfide liposomes) and discuss the reason to use E-100 instead of making cationic liposomes.

Author Response

The authors thank the reviewer for the comments and suggestions made. Likewise, we indicate that the suggested changes will be reported in red to facilitate review.

Reviewers' comments:

Researchers of this manuscript developed new Anti-microbial peptides and used E-100 coated nano-liposomes to improve stability of food.  This is a well written manuscript.

1.    Reviewer comment: How did Eudragit E-100 prevented agglomeration.  If it is a positive-positive repulsion between nano-liposomes, then why can’t researchers make cationic liposomes instead of coating liposomes with E-100.  Please cite (In-vitro and in-vivo tumor growth inhibition by Glutathione disulfide liposomes) and discuss the reason to use E-100 instead of making cationic liposomes.

Author Response:  The reviewer's comment is very consistent if only the electrostatic stabilization effect is considered. However, we had previously established some considerations before conducting this study, which were:

1. It has been reported that micellar and cationic liposomal systems alone (non-coated) have associated intrinsic toxicity problems[1–4].

2. It has been described that polymer coated liposome systems have greater physicochemical stability and better performance than non-coated liposomes [5–8].

3. Currently, there are very few studies focused to evaluate different properties (physicochemical, biological, etc.) of coated cationic liposomes and even more with the Eudragit E-100 polymer.

For these reasons, we chose to work with a coated cationic liposome and not with a non-coated cationic liposome (NCL). Besides, we have been working with liposomes coated with Eudragit:   

Arévalo, L.M.; Yarce, C.J.; Oñate-Garzón, J.; Salamanca, C.H. Decrease of Antimicrobial Resistance through Polyelectrolyte-Coated Nanoliposomes Loaded with β-Lactam Drug. Pharmaceuticals 2019, 12, 1. https://www.mdpi.com/1424-8247/12/1/1

On the other hand, the authors agreed that it is necessary to establish a comparison between the non-coated cationic liposomes and the cationic coated liposomes. Currently, our researches are focused on several goals, such as:

1. To compare the antimicrobial effect on the strains Listeria monocytogenes and Escherichia coli, using the +2 and +5 peptides loaded in several types of cationic liposomes.

2. To compare the antimicrobial effect on the strains Listeria monocytogenes and Escherichia coli, using the peptides +2 and +5 loaded in liposomes coated with polymers derived from chitosan.

3. Compare the physicochemical stability of cationic liposomal systems (non-coated) versus liposomes coated with chitosan.

Therefore, we hope to provide throughout this year a new manuscript focused on to provide information about the performance of two types of liposomes.

Regarding to liposomal electrostatic stabilization, we should comment that the stabilizing effect given by cationic liposomes (Non-coated-liposome-NCL) are different than those conducted by polymer-coated liposomes (PCL). In the case of NCL, the stabilization effects are mainly given by: (i) inter-liposomal electrostatic repulsion and (ii) formation of long-distance aggregates (flocculate aggregates). Whereas in the case of liposomes coated with the Eudragit E-100, these liposomes have more stabilizing effects. This effect is due to the Eudragit E-100 polymer has different proportions of substituent groups in the polymer backbone corresponding to approximately 25% of dimethylamino groups (which are positively ionized in acid medium) and 75% of alkyl esterified groups. Therefore, in the case of the PCL, the stabilization effect is providing by electrostatic and steric repulsion. Besides, such polymer could provide an extra stabilization by means of rheological control (formation of structured vehicle), due to this polymer has a molecular weight of ~48,000 Dalton.

To conclude, the authors consider that the Reviewer´s recommendation is very interesting and successful. For this reason, we included in the current manuscript a brief comparative description between the coated cationic liposome versus non-coated cationic liposomes. Likewise, we include the recommended reference. The information included is located in: page 12, lines 375-380.

Reviewer 2 Report

S. Cantor et al. has reported the physicochemical features of two antimicrobial peptides (peptide +2 and +5), and the polymer-coated nanoliposomes loaded the peptide by molecular dynamic simulation. They also evaluated the antibiological effects of synthesized the peptides and liposomes on Gram-negative and Gram-positive bacteria related to food phathogens. This work has strongly suggested the effective and safety bacteria-control methods composed of the synthesized peptide and cationic-polymer coated nanoliposomes. However, there are some questions and comments that help the readers to understand precisely. 1. Line 73, is “nisin” same word to Nisin? 2. Line 148 and 212, please write the software company, the city, the country (region). 3. Line 155, what is the buffer? 4. Line 174, please show the city and country of “Eppendorf”. 5. Line 195, what is MHB? 6. Table 1, please adjust the position of the sequence number (1, 10, 20). 7. Figure 3B, please change the legend of arginine from ARG to Arg (or Arg4). 8. Figure 5, what were the asterisk(*)marked samples compared to? 9. Is the illustration displayed on the line 307 Figure 5D? 10. line 313–324, please check the meaning of sentences and the legends. Ex.) Was NCL-peptide +2 only increased the size? (line 315) In line 316, was Figure 2B correct? Similarly, were +3 (line 317) was Figure 3B (line 320) correct?

Author Response

The authors thank the reviewer for the comments and suggestions made. Likewise, we indicate that the suggested changes will be reported in red to facilitate review.

Reviewers' comments:

S. Cantor et al. has reported the physicochemical features of two antimicrobial peptides (peptide +2 and +5), and the polymer-coated nanoliposomes loaded the peptide by molecular dynamic simulation.

1. Line 73, is “nisin” same word to Nisin?

Author response: We agree with the comment and have made the changes in line 73.

2. Line 148 and 212, please write the software company, the city, the country (region).

Author response: We agree with the comment and have made the changes in lines 150 and 226

3. Line 155, what is the buffer?

Author response: The comment has been corrected in line 157-158.

4. Line 174, please show the city and country of “Eppendorf”.

Author response: The comment has been corrected in line 178.

5. Line 195, what is MHB?

Author response: it is Mueller Hinton Broth and it Is definite in line 95-96.

6. Table 1, please adjust the position of the sequence number (1, 10, 20).

Author response: the change was made.

7. Figure 3B, please change the legend of arginine from ARG to Arg (or Arg4).

Author response: The comment has been corrected in the manuscript.

8. Figure 5, what were the asterisk(*)marked samples compared to?

Author response: Significant difference P<0.05 to the NCL and CL without peptide (NCL-F and CL-F, respectively)

9. Is the illustration displayed on the line 307 Figure 5D?

Author response: yes, the figure SD is the schematic representation. So, the change was made.

10. line 313–324, please check the meaning of sentences and the legends. Ex.) Was NCL-peptide +2 only increased the size? (line 315) 

Author response: Is described that both peptides increased the size of the liposome in line 333-334.

In line 316, was Figure 2B correct? 

Author response: The comment has been corrected in line 335.

Similarly, were +3 (line 317) 

Author response: The comment has been corrected in line 336.

was Figure 3B (line 320) correct?

Author response: The comment has been corrected in line 339.

Reviewer 3 Report

I think it is an interesting and nicely written article, however there are minor issues that should be corrected. Hence, I recommend publishing this article after minor corrections.

One of the most important things is that MIC results cannot be compared using ANOVA, because it is semiquantitative, not quantitative method. The results should be expressed either as a single value (as it is done in this article) or if there are variations between the repetitions they should be expressed as a range (for example if approximately half of the repetitions gave MIC 1uM, the other half: 2 uM, the MIC should be expressed as a range1-2 uM, not as average+/-SD. The MIC values are considered as different if they vary by more than one dilution, so for example values 1 and 2 are not different, but 1 and 4 are different (and if we have the range1-2 it is different than 4). So in the section 3.6 please delete (p<0.05) (I think they were lines 374 and 409). Also, MIC values 62.5 and 125 uM (line 391) are not different, they can be considered as the same (only 1 dilution difference), so please verify the interpretation 

Other corrections:

line 55: do you have information about activity of this peptide against Gram-positive bacteria or other microorganisms like fungi?

lines 58-60: if the aim is food preservation, the bioavailability is not necessary, moreover it is beneficial that the peptide is not abosrbed and does not exert pharmacological/toxic effect

Methods:

2.5.1-please, provide the details how/on what the drop was depisited

line 155: which buffer? PBS? please provide composition

line 158: delete /L: uM is sufficient; what was the medium? water or buffer?

line 174: did you use Amicon Ultra-0.5 ml centrifugal filter devices? was the supplier Eppendorf (please, provide company, city, state). 

How were the liposomes recovered after this filtration process?

line 176 pH4- which acid or buffer did you use? HCl? acetic acid?

line 198: diluted with what? medium?

line 203: did you measure the turbidity or did you determine the presence/absence of growth visually?

Results: 

the lack of surface activity of peptides in water is surprising. First of all, I think you tested to low concentrations. Also, I don't know how the samples were prepared, I mean in which vessels (glass?) was the drop deposited on a glass surface? Small peptides are strongly adsorbed on different surfaces (glass, filters), there is saturation effect, so this phenomenon is very important at low peptide concentration, but negligible at high concentration. The adsorption of peptide from water is higher that from media with high ionic strength such as PBS. That may also be responsible for the capability of decreasing surface tension in PBS that was not observed for water.

Regarding the DLS results, the polydispersity values 0.5-0.7 are quite high, so if the results are not good quality (you have quality report in the instrument) they should be interpreted with caution. Also, if you have different size populations (line 329), the mean size is meaningless, in this case it would be better to show the graphs with size distribution for comparison.

line 317: did you mean +3 or +5?

line 323: interacts with the surface (please, delete the word 'area')

line 324-336: Figure 5 suggests that the peptide is not located in the lamellar structure, but in the internal aqueous compartment of liposome. Please, rephrase

lines 337-344: I strongly agree that Bradford colorimetric assay is not appropriate in this case. It has a lot of interactions not only with Eudragit, but also with lipids...I don't think mass spectrometry is the best choice, did you mean liquid chromatography-mass spectrometry (LC-MS)? Apart from MS detector, you could also use HPLC with UV-Vis deterctor, it works well with the peptides like yours. 

Since you 'indirectly suggest the vehiculisation of the peptide in liposomes' you should add that you don't know what is the fraction of the peptides vehiculised/associated with the liposomes? it may be 30%, but also 90%- you would need to determine it to be sure, you can't conclude it based on the size/zeta potential measurements. 

Personally I think that even if there is an important fraction of free peptide, there is no need to separate it from liposomes, because such association is an equilibrium process (i.e. if you remove the free peptide, some of the bound peptide will be immediately released). Also, for the antibacterial effect the fraction of weakly bound peptide or free peptide may be beneficial, because if the peptide is very strongly associated with the nanocarrier, if the affinity to nanocarrier is much higher that that for bacteria, it will loose its antibacterial effect. It is important that the bacteria are exposed to lethal peptide concentration, because if the concentration is sublethal, it may give them the opportunity to adapt and acquire resistance.

lines 398/399 please verify carefully reference 39, because plectasin and its derivatives, although they are antimicrobial peptides, are very different from 'classical' antimicrobial peptides, because their target is cell wall, not cell membrane. Compared with your peptides, plectasins have different spectrum (they are active only against gram positive bacteria, not against gram negative), different structure. I would delete this sentence.

Figures 7,8: MIC is sufficient, you can delete B (activity increase)

line 412 not clear- did you mean that blank coated liposomes (i.e. liposomes without peptide) also exhibited the antibacterial activity?

lines 423-426: please, rephrase, not clear

Author Response

The authors thank the reviewer for the comments and suggestions made. Likewise, we indicate that the suggested changes will be reported in red to facilitate review.

Reviewers' comments:

I think it is an interesting and nicely written article, however there are minor issues that should be corrected. Hence, I recommend publishing this article after minor corrections.

1.    Reviewer comment: One of the most important things is that MIC results cannot be compared using ANOVA, because it is semiquantitative, not quantitative method. The results should be expressed either as a single value (as it is done in this article) or if there are variations between the repetitions they should be expressed as a range (for example if approximately half of the repetitions gave MIC 1uM, the other half: 2 uM, the MIC should be expressed as a range1-2 uM, not as average+/-SD. The MIC values are considered as different if they vary by more than one dilution, so for example values 1 and 2 are not different, but 1 and 4 are different (and if we have the range1-2 it is different than 4). So, in the section 3.6 please delete (p<0.05) (I think they were lines 374 and 409). Also, MIC values 62.5 and 125 uM (line 391) are not different, they can be considered as the same (only 1 dilution difference), so please verify the interpretation

Author response: We agree and made the suggested change. However, line 391 (previously report) was unmodified because the results were reproducible by three times. Besides, we followed the Clinical and Laboratory Standards Institute (CLSI) standard, where MIC is defined as "the lowest concentration of antimicrobial agent that completely inhibits growth of the organism in the tubes or microdilution wells as detected by the unaided eye…

Clinical and Laboratory Standards Institute (CLSI), CLSI M07-A9 Methods for Dilution Antimicrobial Susceptibility Tests for Bacteria That Grow Aerobically ; Approved Standard — Ninth Edition, 2012. doi:10.1039/c1jm10248f.

2.    Reviewer comment: line 55: do you have information about activity of this peptide against Gram-positive bacteria or other microorganisms like fungi?

Author response: The activity of the peptide Alyteserin 1c against S. aureus ATCC 25923 has been reported by two studies, but there are no reports of the activity against fungi.

Conlon et al 2009 reported a MIC of 200 µM. Conlon, J.M.; Demandt, A.; Nielsen, P.F.; Leprince, J.; Vaudry, H.; Woodhams, D.C. The alyteserins: two families of antimicrobial peptides from the skin secretions of the midwife toad Alytes obstetricans (Alytidae). Peptides 2009, 30, 1069–1073.

3.    Reviewer comment: lines 58-60: if the aim is food preservation, the bioavailability is not necessary, moreover it is beneficial that the peptide is not abosrbed and does not exert pharmacological/toxic effect

Author response: The term "bioavailability" in line 60 (previously report), is not directly related to a pharmacological effect, but with an environmental approach. In this phrase, refers to "bioavailability" to the peptide must retain its physicochemical and structural properties in an environment, in order to exert the biological effect (antimicrobial).

Regarding to Methods:

4.    Reviewer comment: please, provide the details how/on what the drop was deposited.

Author response: the pendant drop methodology was used to determine the surface tension of the peptide aqueous solutions. For this, the solution was placed into a glass syringe provided by the equipment used. Attached link of the manufacturer: https://www.dataphysics-instruments.com/products/oca/

Besides, we indicate in the manuscript that the pendant drop method was used to determinate the surface tension and two new references were added according to this methodology. (page 4, line 156).

5.    Reviewer comment: line 155: which buffer? PBS? please provide composition

Author response: We agree with the comment and have made the changes. (page 4, lines 157-158), where we indicated that the used buffer was PBS pH 7.2 and 138 mM of ionic strength.

6.    Reviewer comment. line 158: delete /L: uM is sufficient; what was the medium? water or buffer?

Author response: We agree with the comment and have made the changes. (page 4, line 161)

7.    Reviewer comment: line 174: did you use Amicon Ultra-0.5 ml centrifugal filter devices? was the supplier Eppendorf (please, provide company, city, state).

Author response:  We used centrifugal filter tubes, provided by VWR™ with a 30 kDa of cut-off.  We have made the correction in the manuscript with a better explanation of the liposomes formation process. This is indicated in page 4, lines 174-188.

8.    Reviewer comment: How were the liposomes recovered after this filtration process?

Author response: We have made the correction in the manuscript with a better explanation of the liposomes formation process. This is indicated in page 5, lines 174-188.

9.    Reviewer comment: line 176 pH4- which acid or buffer did you use? HCl? acetic acid?

Author Response:  We have made the correction in the manuscript with a better explanation of the liposomes modification process. This is indicated in page 5, lines 190 to 195.

10.  Reviewer comment: line 198: diluted with what? Medium?

Author Response:  We rearrange the paragraph providing the suggested information in page 5, lines 190 to 195.

11. Reviewer comment: line 203: did you measure the turbidity, or did you determine the presence/absence of growth visually?

Author Response:  we determinate the presence/absence of growth visually. This information was included in page 6, lines 218-219

Regarding to Results:

12.  Reviewer comment: the lack of surface activity of peptides in water is surprising. First of all, I think you tested to low concentrations. Also, I don't know how the samples were prepared, I mean in which vessels (glass?) was the drop deposited on a glass surface? Small peptides are strongly adsorbed on different surfaces (glass, filters), there is saturation effect, so this phenomenon is very important at low peptide concentration, but negligible at high concentration. The adsorption of peptide from water is higher that from media with high ionic strength such as PBS. That may also be responsible for the capability of decreasing surface tension in PBS that was not observed for water.

Author Response:  We agree with the comment and we would like to clarify some interesting aspects. First of all, the laboratory supplies used for sample preparation were manufacturing with polypropylene and some of them were Nalgene™ brand. Secondly, in the case of surface tension measurements, we used the pendant drop method because this method allows us to determine the surface activity in the peptides aqueous solution without its deposition over a solid surface; this is also called as drop shape analysis. Doctors, Sameh M.I, Saad A and Wilhelm Neumann1 states in their manuscript:

“… Drop shape techniques for the measurement of interfacial tension are powerful, versatile and flexible. They need only small amounts of liquid and do not depend on any knowledge of other parameters, such as the contact angle, or on an empirical factor. Since the volume of a drop, possibly suspended from the end of a capillary, can be easily changed, changes in surface area will occur concomitantly, hence offering an approach to run a drop shape method as a surface film balance. In such a setup, liquid-vapour interfacial tensions as small as 1.0 mN/m and lower can be measured accurately. There is indeed no lower limit of surface tension for such measurements, and values of 10−3 and lower have been reported for liquid-liquid systems. There is also no upper limit on surface tension for the applicability of drop shape techniques…”

Sameh M.I. Saad A. Wilhelm Neumann. Axisymmetric Drop Shape Analysis (ADSA): An Outline. Advances in Colloid and Interface Science. Volume 238, December 2016, Pages 62-87. https://doi.org/10.1016/j.cis.2016.11.001

However, the consideration that the peptide could have been adsorbed on the wall of the equipment syringe, is a possibility that should also be considered. Therefore, the authors decided to include them in the current manuscript. (page 9, lines 302-306).

13.  Reviewer comment: Regarding the DLS results, the polydispersity values 0.5-0.7 are quite high, so if the results are not good quality (you have quality report in the instrument) they should be interpreted with caution. Also, if you have different size populations (line 329), the mean size is meaningless, in this case it would be better to show the graphs with size distribution for comparison.

Author Response:  We agree with the comment and we believe that is a very interesting topic which is worthy to be addressed. Initially, for the liposomes without the coating process, it was obtained an adequate value of polydispersity (PDI<0.3); whereas with coated liposomes the PDI increases to values of 0.5 – 0.7, both cases with good quality measurements according to the Zsizer nano ZSP software. When we indicated that probably there is liposomes with different size populations adhering to the hypothesis that the coating polymer interact in different ways to the liposomal surface, we did not obtain a graph with different population distribution showing several peaks (i.e, bimodal distribution). In fact, we observed an increment in the distribution range and due to this we have worked with the mean size always indicating the error bars in the result. Nevertheless, we are currently working in a paper which will allow clarifying these interactions regarding the association modes between this coating polymers and liposomes with different surface properties.

14.  Reviewer comment: line 317: did you mean +3 or +5?

Author Response:  We agree and made the suggested change.

Reviewer comment: line 323: interacts with the surface (please, delete the word 'area')

Author Response:  We agree and made the suggested change.

15.  Reviewer comment: line 324-336: Figure 5 suggests that the peptide is not located in the lamellar structure, but in the internal aqueous compartment of liposome. Please, rephrase

Author Response:  Regarding this, the authors believe that the peptide could be located in several parts of the liposomal system. This can be seen with the changes in size and zeta potential given when the liposome is alone and subsequently when it is loaded with the peptides.

In relation to the change in size, it could be explained by the loading of the peptide into the liposome. However, the decrease in the negative value of the zeta potential on the liposomal surface suggests that the peptides could also be located in the lamellar part, where the positive side of the peptide is towards the medium. Therefore, the authors consider that the current schematic representation shows the possible places where the peptide could be located. So, we prefer keep it.

16. Reviewer comment: lines 337-344: I strongly agree that Bradford colorimetric assay is not appropriate in this case. It has a lot of interactions not only with Eudragit, but also with lipids...I don't think mass spectrometry is the best choice, did you mean liquid chromatography-mass spectrometry (LC-MS)? Apart from MS detector, you could also use HPLC with UV-Vis deterctor, it works well with the peptides like yours.

Author Response:  We agree and made the suggested change in line 362-366.

17. Reviewer comment: Since you 'indirectly suggest the vehiculisation of the peptide in liposomes' you should add that you don't know what is the fraction of the peptides vehiculised/associated with the liposomes? it may be 30%, but also 90%- you would need to determine it to be sure, you can't conclude it based on the size/zeta potential measurements.

Author Response:  We agree with this interesting comment and made the suggested change in lines 362 to 366. Also, we want to indicate that our laboratory has been worked in this field:

1.       Lina M. Arévalo, Cristhian J. Yarce, José Oñate-Garzón and Constain H. Salamanca. Decrease of Antimicrobial Resistance through Polyelectrolyte-Coated Nanoliposomes Loaded with β-Lactam Drug. Pharmaceuticals 2019, 12(1), 1; https://doi.org/10.3390/ph12010001

2.       Constain H. Salamanca, Cristhian J. Yarce, Yony Roman, Andrés F. Davalos and Gustavo R. Rivera. Application of Nanoparticle Technology to Reduce the Anti-Microbial Resistance through β-Lactam Antibiotic-Polymer Inclusion Nano-Complex. Pharmaceuticals 2018, 11(1), 19; https://doi.org/10.3390/ph11010019

18. Reviewer comment: Personally, I think that even if there is an important fraction of free peptide, there is no need to separate it from liposomes, because such association is an equilibrium process (i.e. if you remove the free peptide, some of the bound peptide will be immediately released). Also, for the antibacterial effect the fraction of weakly bound peptide or free peptide may be beneficial, because if the peptide is very strongly associated with the nanocarrier, if the affinity to nanocarrier is much higher than that for bacteria, it will lose its antibacterial effect. It is important that the bacteria are exposed to lethal peptide concentration, because if the concentration is sublethal, it may give them the opportunity to adapt and acquire resistance.

Author Response:  We completely agree with your point of view..

19. Reviewer comment: lines 398/399 please verify carefully reference 39, because plectasin and its derivatives, although they are antimicrobial peptides, are very different from 'classical' antimicrobial peptides, because their target is cell wall, not cell membrane. Compared with your peptides, plectasins have different spectrum (they are active only against gram positive bacteria, not against gram negative), different structure. I would delete this sentence.

Author Response:  Sentence was deleted.

20. Reviewer comment: Figures 7,8: MIC is sufficient, you can delete B (activity increase)

Author Response:  We considered it is relevant to express the activity in “times”, especially in L. monocytogenes where liposomes strongly contributed to the antibacterial effect.

21.  Reviewer comment: line 412 not clear- did you mean that blank coated liposomes (i.e. liposomes without peptide) also exhibited the antibacterial activity?

Author Response:  Yes. The sentence was cleared in line 436.

22. Reviewer comment: lines 423-426: please, rephrase, not clear

Author Response:  The sentence was cleared in lines 448-449.